# Analysis of Spinal Ischemia after Frozen Elephant Trunk for Acute Aortic Dissection: An Observational, Single-Center Study

**DOI:** 10.3390/diagnostics12112781

**Published:** 2022-11-14

**Authors:** Frederico Lomonaco Cuellar, Alexander Oberhuber, Sven Martens, Andreas Rukosujew, Elena Marchiori, Abdulhakim Ibrahim

**Affiliations:** 1Department of Vascular and Endovascular Surgery, University Hospital Muenster, 48149 Muenster, Germany; 2Department of Cardiothoracic Surgery, Division of Cardiac Surgery, University Hospital Muenster, 48149 Muenster, Germany

**Keywords:** frozen elephant trunk, spinal cord ischemia, aortic dissection

## Abstract

Background: This observational study aimed to evaluate the perioperative risk factors for spinal cord ischemia (SCI) in patients who underwent aortic repair with the frozen elephant trunk technique (FET) after acute aortic Stanford A dissection. Methods: From May 2015 to April 2019, 31 patients underwent aortic arch replacement with the FET technique, and spinal ischemia was observed in 4 patients. The risk factors for postoperative SCI were analyzed. Results: The mean age of patients with acute aortic dissection was 57.1 years, and 29.4% were female. Four patients developed SCI. There were no significant differences in characteristics such as age and body mass index. The female gender was associated with most of the SCI cases in the univariate analysis (75%, *p* = 0.016). Known perioperative and intraoperative risk factors were not related to postoperative SCI in our study. Patients who developed SCI had increased serum postoperative creatinine levels (*p* = 0.03). Twenty-four patients showed complete false lumen thrombosis up to zones 3–4, five patients up to zones 5–6 and two patients up to zones 7–9, which correlates with the postoperative development of SCI (*p* = 0.02). The total number of patent intercostal arteries was significantly reduced postoperatively in SCI patients (*p* = 0.044). Conclusions: Postoperative acute kidney injury, the reduction in patent intercostal arteries after surgery and the extension of false lumen thrombosis up to and beyond zone 5 may play a significant role in the development of clinically relevant spinal cord injury after FET.

## 1. Introduction

The conventional treatment for acute type A aortic dissection is the replacement of the ascending portion of the aorta, which eliminates the primary entry tear. The residual false lumen may involve the entire downstream aorta and plays an essential role in determining the prognosis [1]. As long as the false lumen remains patent, the chances of survival might remain in jeopardy, and reoperation is inevitable. Therefore, it is expected that the long-term prognosis may be improved by performing a procedure that obliterates the residual false lumen in the descending thoracic aorta when treating acute type A aortic dissection. The frozen elephant trunk technique (FET) combines total aortic arch replacement with the intraoperative stenting of the distal thoracic aorta. Its purpose of achieving the complete thrombosis of the false lumen in the descending aorta may reduce the necessity of further operations to treat the residual false lumen [2] and provides more favorable remodeling in the descending aorta than other methods [3].

Although these favorable features of the FET procedure are promising, the incidence of postoperative spinal ischemia (SCI) is a matter of concern. The rate of postoperative SCI has been reported as high as 6.8% in patients after hybrid aortic repair [4] and 4% in patients after FET for acute type A aortic dissection [5]. The purpose of our study is to evaluate the risk factors for SCI in patients after aortic arch repair using FET for Stanford A dissection.

## 2. Materials and Methods

From May 2015 to April 2019, 31 patients underwent aortic arch replacement with the frozen elephant trunk (FET) technique using a Thoraflex Hybrid prosthesis. Spinal ischemia was observed in 4 patients. Risk factors for SCI in this group of 31 patients were evaluated in a retrospective analysis. The number of patent intercostal arteries, the dominance of the left vertebral artery, the occlusion or dissection of the subclavian artery and iliac arteries, and the extension of false lumen thrombosis were analyzed in preoperative and postoperative CT scans. Known risk factors for postoperative SCI, such as perioperative hemoglobin level, hypoxemia, hypotension, and increases in serum creatinine levels, were measured.

Surgery was performed under hypothermic (28 °C) circulatory arrest and selective bilateral antegrade cerebral perfusion. Near-infrared spectroscopy (NIRS) was used to monitor cerebral tissue oxygenation. After median sternotomy, extracorporeal circulation was initiated. Myocardial protection was achieved using retrograde cold blood cardioplegia. After the distal transection of the aortic arch, the FET was applied. A Thoraflex™Hybrid Plexus was positioned in the descending aorta. The stent graft size was determined according to the maximal diameter of the true lumen. After the completion of the distal anastomosis in zone 3, aortic arch vessel reconstruction was performed under reinitialized perfusion of the lower body.

Continuous variables were expressed as means ± standard deviations for parametric data and medians with interquartile ranges for non-parametric data, whereas dichotomous variables were reported as crude numbers and percentages. Comparisons of continuous variables were performed using a Student’s *t*-test for normally distributed variables and a Mann–Whitney U test for non-normally distributed variables. A chi-square test or Fisher’s exact test was used for categorical variables. A *p*-value of <0.05 was considered statistically significant. All statistical analyses were performed using SPSS Statistics for Windows, version 26.0 (IBM Corp., Armonk, NY, USA).

## 3. Results

The mean age of patients with acute aortic dissection was 57.1 (IQR of 55.5–58.7) years, and 29.4% were female. The mean body mass index was 27.7 kg/m^2^ (26.8–28.6). Four patients underwent a concomitant TEVAR distal to FET. Twenty-two had hypertension (70.9%); nine patients were smokers (29%); one had diabetes type 2 (3.2%); two had suffered a prior stroke (6.4%); five had known atrial fibrillation (16.1%). Five patients were under acetylsalicylic acid (ASA) (16.1%), two under anticoagulation with Phenprocoumon (6.4%) and four under statin therapy (12.9%). Four patients developed SCI. Three of them were women. One female patient suffered from tetraplegia (Tarlov scoring scale = 0), with sensory loss up to the eighth thoracic spinal nerve; her postoperative MRI scan showed a hyperintense spinal cord lesion from the third to the tenth thoracic vertebrae; cerebrospinal fluid drainage was not performed because of hemodynamic instability. One female patient developed paraplegia on the second postoperative day (Tarlov scoring scale = 0), which recovered completely after cerebrospinal fluid drainage. One female patient developed paraparesis on the first postoperative day (Tarlov scoring scale = 3), which recovered completely on the left side, but no recovery on the right leg after cerebrospinal fluid drainage was observed. The male patient suffered from paraplegia on the first postoperative day (Tarlov scoring scale = 0), which did not improve after cerebrospinal fluid drainage; none of them underwent concomitant TEVAR; all of them underwent concomitant aortic valve replacement.

All 31 patients underwent total arch replacement with FET. Twenty patients received an FET of 100 cm in length; eleven patients received an FET of 150 mm in length.

The data of all patients were divided into two groups: patients with SCI and without SCI. There were no significant differences in characteristics such as age (SCI and non-SCI, 62.5 (59.5–65.5); *p* = 0.37), body mass index (SCI, 24.6 kg/m^2^ (23.94–25.26) vs. non-SCI, 27 kg/m^2^ (27.98–26.02)). Female gender was, however, associated with most of the SCI cases in the univariate analysis, but not in the multivariate analysis (75%, *p* = 0.016). Furthermore, the perioperative use of ASA, betablockers, ACE inhibitors, anticoagulants and statins and known essential hypertension, COPD, atrial fibrillation, coronary heart disease, previous heart surgery, heart insufficiency, diabetes and prior stroke or transitory ischemic attack were not associated with SCI. Preoperative CT scans showed unilateral renal ischemia in one SCI patient, which also did not show relation to SCI (Table 1).

Intraoperative cardiopulmonary bypass time, hypothermic circulatory arrest time, aorta cross-clamping times, concomitant replacement of the aortic valve and cardiac revascularization were not related to postoperative SCI. Likewise, the intraoperative transfusion amounts of erythrocytes, platelets, fibrinogen and prothrombin complex concentrate were not related to postoperative SCI; the length of the stent graft (100 compared to 150 mm) was not related to postoperative SCI (Table 2).

Patients who developed SCI had increased postoperative serum creatinine levels (SCI, 1.0 mg/dL (0.94–1.06) vs. non-SCI, 0.66 mg/dL (0.58–0.74); *p* = 0.032). All SCI patients fulfilled the AKIN criteria for acute kidney disease (AKI) [6]. The perioperative hemoglobin levels were not associated with SCI (SCI, 11.3 mg/dL (9.8–12.8) vs. non-SCI, 13.5 mg/dL (13.2–13.8); *p* = 0.29), nor was the perioperative platelet count (SCI, 288 × 1000/μL (405.6 × 1000/μL–254.8 × 1000/μL) vs. non-SCI, 188 × 1000/μL (175.6 × 1000/μL–200.4 × 1000/μL); *p* = 0.472). Hypotension (preoperative, intraoperative and postoperative mean values under 65 mmHg), hypoxemia (arterial partial pressure of oxygen under 75 mmHg), postoperative hemoglobin values and platelet count did not show associations with the development of SCI. Hypotension (preoperative, intraoperative and postoperative mean values under 65 mmHg), hypoxemia (arterial partial pressure of oxygen under 75 mmHg), postoperative hemoglobin values and platelet count did not show associations with the development of SCI. Left-sided ischemic stroke was observed in the postoperative CT scan of one SCI patient. Both findings showed no statistical significance. The extension of postoperative false lumen thrombosis was analyzed according to the SVS and STS reporting standards [7]. Postoperative complete false lumen thrombosis from zone zero to zone five was observed in all four of the SCI patients. One patient showed complete false lumen thrombosis up to zone 5, one patient up to zone 6, one patient up to zone 7 and one patient up to zone 9. Among non- SCI patients, nineteen showed complete false lumen thrombosis up to zone 4, five patients up to zone 3 and three patients up to zone 5. The postoperative extension of false lumen thrombosis after FET was associated with development of SCI (*p* = 0.002) (Table 3).

The preoperative and postoperative patency of the intercostal arteries is shown in Figure 1. There was no statistically significant difference in the number of total patent intercostal arteries prior to surgery, including those originated not only from the true lumen but also from the false lumen, between the two groups. (SCI, 20.5 (18.4–22.6) vs. non-SCI, 22 (19.8–24.2); *p* = 0.52). However, the reduction in the number of total patent intercostal arteries after FET was associated with postoperative SCI (SCI, 14.5 (10.4–18.5) vs. non-SCI, 20.5 (17.2–23.8); *p* = 0.04). The postoperative reduction in the number of patent intercostal arteries originated from the true lumen showed no statistical significance (SCI preoperative, 13 (7.4–18.6) vs. non-SCI preoperative, 12 (6.2–17.8); *p* = 0.88; SCI postoperative, 7 (4.3–9.7) vs. non-SCI postoperative, 7 (1.7–12.3); *p* = 0.66). Likewise, the postoperative reduction in patent intercostal arteries originated from the false lumen showed no statistical significance (SCI preoperative, 7 (4.2–9.8) vs. non-SCI preoperative, 11 (5.2–16.8); *p* = 0.41; SCI postoperative, 6 (0.3–11.7) vs. non-SCI postoperative, 12 (5.7–18.3); *p* = 0.13). The Figure 2 shows aortic dissection with aortic zones according to Ishimaru classification (0–11) [7]. Figure 3 (CT scan) shows occlusion of the false lumen and thrombosis of the intercostal arteries.

In total, 1 SCI (25%) and 15 non-SCI patients (55.5%) showed dominance of the left vertebral artery in the CT scan, which was not statistically correlated with postoperative SCI (*p* = 0.275) (Table 3).

## 4. Discussion

The frozen elephant trunk technique has proven itself as a reliable method in acute type A aortic dissection by managing the false lumen in the proximal descending thoracic aorta, a segment known to evolve quickly following conventional surgery [8,9,10]. In a primary intervention, the FET technique aims to depressurize and induce the thrombosis of the dilated persistent false lumen. The FET was initially designed to treat aortic arch and proximal descending thoracic aortic disease as a one-stage procedure. Its uses have evolved into providing a platform for second-stage open and thoracic aortic endovascular repair (TEVAR) procedures, in acute and chronic aortic dissection. A potential shortcoming of the FET is the necessity of a complex arch replacement; a hemiarch replacement might be a more adequate approach in selected cases [11].

An alternative hybrid method for treating acute type A dissection is the modified frozen elephant trunk (mFET). This procedure leaves the native aortic arch in situ with the remaining dissection; the ascending aorta is replaced in open surgery, and the descending aorta is repaired with an endovascular stent graft. In a propensity-based analysis, Berdajs et al. compared the mFET to isolated ascending aorta and hemiarch replacement. Their findings regarding midterm outcomes favored the mFET [12]. However, this study did not compare the outcomes of the mFET and the FET [11].

SCI is a known postoperative complication of the FET technique; this incidence has been reported from 0 to 24% [13,14,15]. In a pooled dataset of 50 studies, the rate of SCI in patients undergoing hybrid aortic arch repair was 6.8%. Furthermore, the FET for acute type A aortic dissection was associated with a postoperative SCI rate of 4%. In their study of 31 patients, Hori et al. reported 4 cases of postoperative SCI (12.9%) [3]. In our analysis of 31 patients, 4 developed postoperative SCI after FET for acute type A dissection.

The deliberate use of the FET has been discouraged by some surgeons because of the increasing evidence suggesting a higher SCI rate after FET than after conventional ascending aorta and hemiarch replacement, even in comparison to the conventional elephant trunk technique. The multicenter ARCH registry found no statistical differences between the rates of SCI after FET and after conventional aortic repair. However, a retrospective ARCH registry analysis by Liakopoulos found several limitations in this registry analysis, proposing that more robust evidence is necessary to determine which subgroup of patients would benefit the most from the FET technique [16].

Several vascular risk factors for SCI have been described and involved diabetes mellitus, hyperlipidemia, atrial fibrillation, previous stroke or transient ischemic attacks, prior myocardial infarction, heart diseases (congestive heart failure, arrhythmia, and valvar heart disease) and cigarette smoking [17]. None of these was related to SCI in our series.

The lowering of the body temperature disrupts the physiological processes at the molecular, cellular and system levels, but hypothermia induced prior to cardiosurgical or neurosurgical procedures by the decrease in tissue oxygen demand can reduce the risk of cerebral or cardiac ischemic damage [18].

Intraoperative risk factors for the development of SCI, such as a core body temperature ≥28 °C during circulatory arrest in combination with a prolonged circulatory arrest time [3], showed no statistical significance in our study.

The extensive vascular collateral network of the spinal cord is provided by the anterior spinal artery, dual posterior spinal arteries and dual posterolateral spinal arteries (Figure 4) [19]. Just a few radicular arteries are responsible for the blood supply to the spinal arteries at the thoracolumbar level. They stem from segmental branches of the aorta (posterior intercostal and lumbar branches), which reach the intervertebral foramina and divide into the anterior and posterior radicular arteries. The largest radicular artery is the arteria radicularis magna of Adamkiewicz (or main anterior radicular artery), which most commonly arises on the left from T9 to T12 but occasionally can be positioned on the right (17% of cases) and arise anywhere from T5 to L4 [20].

The occlusion of intercostal arteries after FET has been reported as a risk factor for SCI by Kozlov et al. [15]. Hori et al. described significant spinal cord perfusion from intercostal arteries originating from the false lumen as a significant risk factor for SCI [3]. In our analysis, patients who developed SCI suffered a relevant postoperative decrease in the number of total patent intercostal arteries, but the number of vessels originating from the false lumen and the postoperative decrease in arteries originated from the true lumen were not statistically related to SCI (Figure 1). Furthermore, the extension of false lumen thrombosis has also been related to the development of SCI. All four SCI patients had a thrombosis extension up to at least zone 5. To this date, there are no reports analyzing the relationship of the extension of false lumen thrombosis after FET and the incidence of postoperative SCI.

Our study also suggests that postoperative acute kidney injury may present a risk factor for the postoperative development of SCI. Renal dysfunction has been previously reported as a risk factor for SCI after aortic repair [3,17].

Several procedures have been suggested to prevent SCI after FET, including keeping the mean blood pressure above 90 mmHg [15]; cooling the patient; decreasing ischemic time during the application of the FET; reducing the overall period of circulatory arrest through distal perfusion by means of LHB circuits or cardiopulmonary bypass with distal renal, visceral and iliac perfusion; and performing cerebral spinal fluid drainage (CSFD) [21,22,23,24,25]. Katayama et al. found no differences in the incidence of SCI after FET between patients with our without CSFD, but in this analysis, only patients at a high risk of postoperative SCI received preoperative CSFD. In our series, no patient received preoperative CSFD. There were no relationships among hypotension (preoperative, intraoperative and postoperative mean values under 65 mmHg), hypoxemia (arterial partial pressure of oxygen under 75 mmHg) and the development of SCI. The female gender was related to postoperative SCI after FET in the univariate analysis but not in the multivariate analysis.

The major limitation of our study is our sample size. Larger sample sizes are required to confirm our results through multivariate analyses. As our study respects a retrospective design, it provides no causal relationship, only a correlational relationship between the status of the intercostal arteries and the development of SCI after FET. Nevertheless, the findings of this study are worth reporting because they might be a significant factor considering when selecting suitable patients for FET. These results should be considered regarding the risks involving postoperative SCI in aortic diseases.

## 5. Conclusions

The frozen elephant trunk is a suitable technique for treating aortic arch and proximal descending thoracic aortic pathology as a one-stage procedure and also for providing a platform for second-stage open and thoracic aortic endovascular repair (TEVAR) procedure and for acute and chronic dissection, positively modifying the natural history of the false lumen. SCI is a known devastating complication of the FET; postoperative acute kidney injury, the total loss of intercostal arteries and the extension of false lumen thrombosis may be risk factors for developing SCI. Further studies are needed to confirm our findings.

## Figures and Tables

**Figure 1 diagnostics-12-02781-f001:**
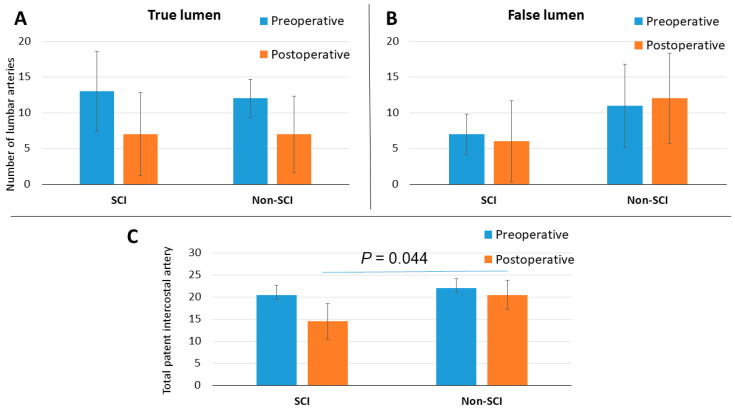
Ratio of patent posterior intercostal arteries before and after FET: (**A**) vessels originated from true lumen; (**B**) vessels originated from false lumen; (**C**) all posterior intercostal arteries.

**Figure 2 diagnostics-12-02781-f002:**
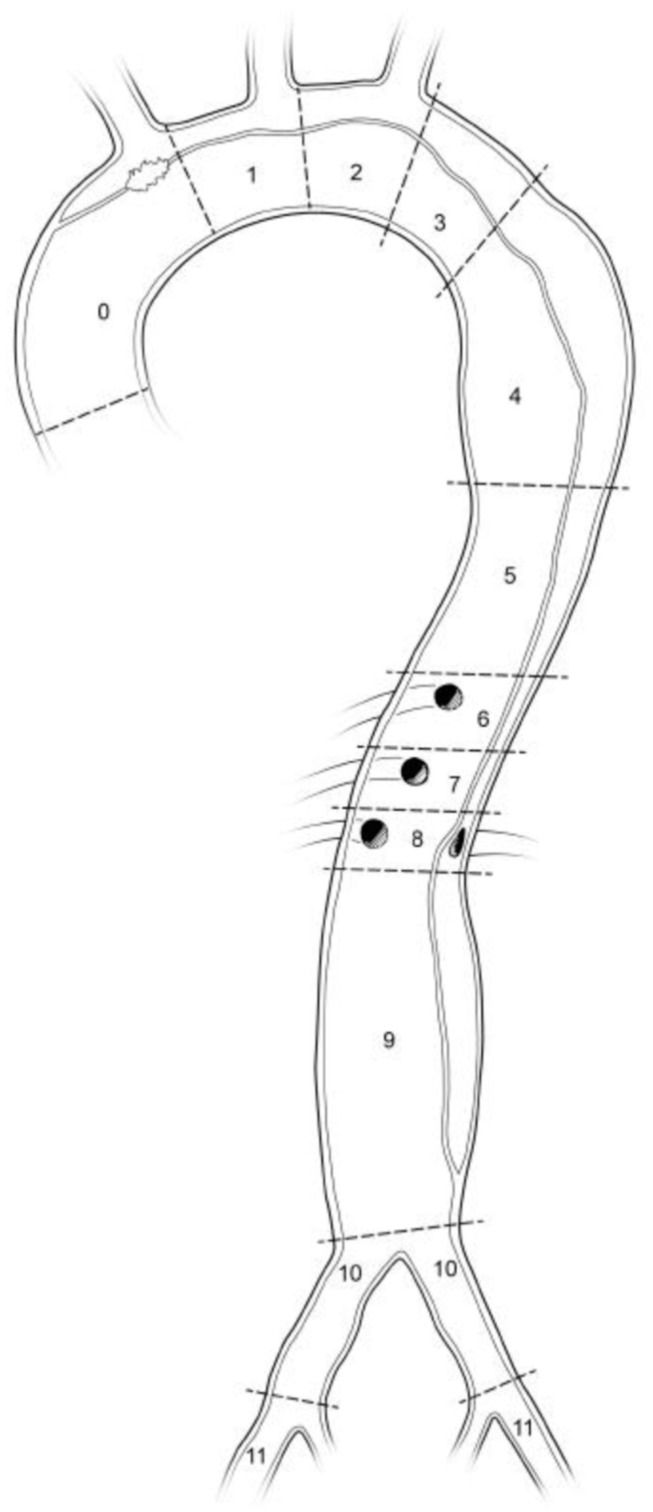
Society for Vascular Surgery/Society of Thoracic Surgeons (SVS/STS) Aortic Dissection Classification System of dissection subtype according to zone location of primary entry tear [7] Ishimaru classification of aortic zones are described from 0 to 11.

**Figure 3 diagnostics-12-02781-f003:**
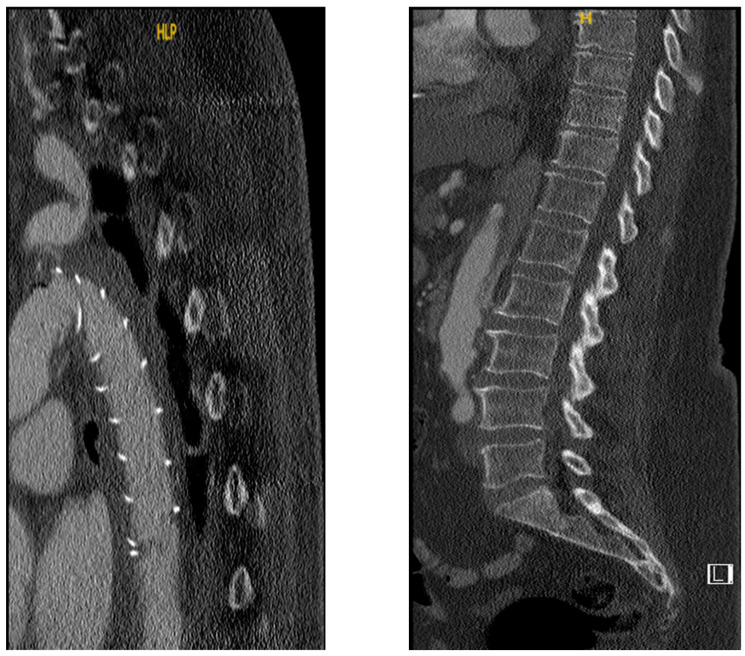
CT scans (sagittal view) of SCI patient. (**Left**): Proximal descending aorta after FET with two visible posterior intercostal arteries without contrast enhancement. (**Right**): Distal aorta of the same patient showing thrombosis of false lumen up to zone 9.

**Figure 4 diagnostics-12-02781-f004:**
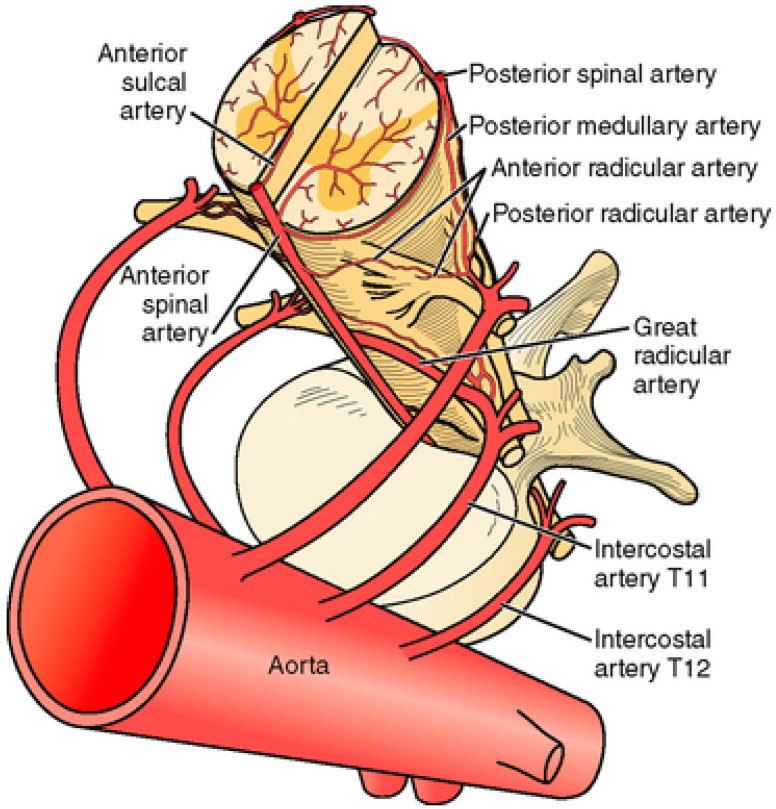
Axial view of the normal vascular anatomy of the spinal cord [20].

**Table 1 diagnostics-12-02781-t001:** Baseline characteristics of the study population.

Variable	Total (*n* = 31)	SCI (*n* = 4)	Non-SCI (*n* = 27)	*p*-Value
Demographic characteristic				
Age (years), mean ± SD	57.1 ± 1.6	62.5 ± 3	62.5 ± 3	0.379
Female gender, *n* (%)	6 (19.3)	3 (75)	3 (11.1)	0.016
BMI (kg/m^2^), mean ± SD	27.7 ± 0.90	24.6 ± 0.66	27 ± 0.98	0.272
Medical history, *n* (%)				
Hypertension	22 (70.9)	2 (50)	20 (74.0)	0.560
Previous stroke/TIA	2 (6.4)	0 (0)	2 (7.4)	0.747
COPD	3 (9.6)	1 (25.0)	2 (7.4)	0.349
Diabetes	1 (3.2)	0 (0)	1 (3.7)	0.871
Coronary artery disease	6 (19.3)	1 (25.0)	5 (18.5)	0.745
Chronic heart disease	2 (6.4)	0 (0)	2 (7.4)	0.755
Current/previous smoker	9 (29.0)	2 (50)	7 (25.9)	0.345
Atrial fibrillation	5 (16.1)	1 (25)	4 (14.8)	0.525
Preoperative cerebral ischemia	0	0	0	0
Postoperative visceral ischemia	5 (16.1)	0 (0)	5 (18.5)	0.589
Medication treatment, *n* (%)				
ß-blocker	12 (38.7)	0 (0)	12 (44.4)	0.121
ACE inhibitor	5 (16.1)	0 (0)	5 (18.5)	0.475
Aspirin	5 (16.1)	1 (25)	4 (14.8)	0.553
Statin	4 (12.9)	0 (0)	4 (14.8)	0.558
Anticoagulation	2 (6.4)	0 (0)	2 (7.4)	0.747

COPD, chronic obstructive pulmonary disease; TIA, transient ischemic attack.

**Table 2 diagnostics-12-02781-t002:** Intraoperative variables.

Variable	Total (*n* = 31)	SCI (*n* = 4)	Non-SCI (*n* = 27)	*p*-Value
CPB time (min) (mean ± SD)	216.4 ± 8.0	209.7 ± 6.4	217,5 ± 9.4	0.743
Aorta clamping time (min) (mean ± SD)	138 ± 9.1	140.7 ± 10.7	137.6 ± 10.5	0.908
Hypothermic circulatory arrest time (min) (mean ± SD)	49.2 ± 1.7	54.5 ± 8.8	48.4 ± 1.5	0.240
CABG, *n* (%)	2 (6.4)	0	2 (7.4)	0.755
Concomitant aortic valve replacement, *n* (%)	20 (64.5)	4 (100)	16(59.2)	0.237
PRBC units (mean ± SD)	9.7 ± 1.1	9 ± 1.0	9.8 ± 1.2	0.809
PC units (mean ± SD)	2.3 ± 0.45	1 ± 0.40	2.5 ± 0.50	0.268
Fibrinogen units (mean ± SD)	5.1 ± 0.44	4.2 ± 0.62	5.3 ± 0.50	0.419
PL units (mean ± SD)	4.6 ± 1.1	2.2 ± 1.3	4.9 ± 1.3	0.452
PCC units (mean ± SD)	4 ± 0.58	2.5 ± 1.04	4.2 ± 0.64	0.291
FET prothesis length 150 mm	11 (0.35)	1 (0.25)	10 (37.0)	0.712

PRBC, packed red blood cell concentrate; PC, platelet concentrate; PL, plasma concentrate. PCC, prothrombin complex concentrate; CABG, coronary artery bypass graft; CPB, cardiopulmonary bypass time.

**Table 3 diagnostics-12-02781-t003:** Postoperative data.

Variable	Total (*n* = 31)	SCI (*n* = 4)	Non-SCI (*n* = 27)	*p*-Value
Laboratory data				
Creatinine (mg/dL)	0.96 ± 0.06	1.0 ± 0.06	0.667 ± 0.08	0.032
Hemoglobin (mg/dL)	13.2 ± 0.34	11.3 ± 1.5	13.5 ± 0.3	0.29
Platelet count (×1000/μL)	199.5 ± 16.73	288 ± 113.6	188 ± 12.4	0.472
Risk factor for SCI, *n* (%)				
Hypoxemia (day 0) (mmHg)	5 (16.1)	1 (25.0)	4 (14.8)	0.553
Hypoxemia (POD 1) (mmHg)	11 (35.4)	2 (50.0)	9 (33.3)	0.336
Hypoxemia (POD 2) (mmHg)	13 (41.9)	3 (75.0)	10 (37.0)	0.124
Hypoxemia (POD 3) (mmHg)	26 (83.8)	3 (75.0)	23 (85.1)	0.454
Hypotension (day 0) (mmHg)	26 (83.8)	3 (75.0)	23 (85.1)	0.454
Hypotension (POD 1) (mmHg)	1 (3.2)	0 (0)	1 (3.7)	0.871
Hypotension (POD 2) (mmHg)	18 (58.0)	2 (50.0)	16 (59.2)	0.452
Hypotension (POD 3) (mmHg)	12 (38.7)	2 (50.0)	10 (37.0)	0.705
Dominating left vertebral artery	16 (51.6)	1(25.0)	15(55.5)	0.275
Postoperative cerebral ischemia	4 (12.9)	1 (0.25)	3 (11.1)	0.429
Postoperative visceral ischemia	9 (29.0)	2 (50)	7 (25.9)	0.429
Thrombosis extension in false lumen, *n* (%)				
Zones 3–4	24 (77.4)	0 (0)	24 (88.8)	
Zones 5–6	5 (16.1)	2 (50.0)	3 (11.1)	
Zones 7–9	2 (6.4)	2 (50.0)	0 (0)	0.002

POD, postoperative day; hypotension, mean arterial pressure under 65 mmHg; hypoxemia, arterial partial pressure of oxygen under 75 mmHg.

## Data Availability

The data used to support the findings of this study are included within the article.

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
