# Peer review of "Analysis of Spinal Ischemia after Frozen Elephant Trunk for Acute Aortic Dissection: An Observational, Single-Center Study"

_diagnostics, 2022, doi:10.3390/diagnostics12112781_

Round 1

Reviewer 1 Report

The authors of the manuscript aimed to assess the perioperative risk factors for SCI in patients who underwent aortic repair with FET technique 12 after acute aortic Stanford A dissection. They included 31 pts of which 4 suffered from SCI. They concluded that postop acute kidney injury, reduction of patent intercostal arteries after surgery and extension of false lumen thrombosis up to and beyond zone 5 may play a significant role in the development of clinically relevant SCI after FET.

Major remarks:

1.      I am pretty sure the study is heavily underpowered thus it’s almost impossible to find any relevant predictors of SCI in such small pts group. This is also the major limitation of the current study.

2.      In such low sample sized studies the authors should emphasize observational form of the study.

3.      Moreover, the statistics reminds comparison of apples with plums and nothing more.

4.      Ask your statistician why UV regression and MV regression could not be performed.

Minor remarks:

1.      In the methods section please specify whether the study was prospective or retrospective

2.      Did you otain IERB approval?

3.      What was the duration of follow-up?

4.      The best feature of  the current manuscript are the figures. Congratulations.

5.      English language requires minor editing.

Author Response

Muenster, 31/10/2022

Editorial Office

MDPI: Diagnostics

Object: ID: diagnostics- 1987419

Analysis of spinal ischemia after frozen elephant trunk for acute aortic dissection: an observational single center study

Dear editors and reviewers,

Thank you for your feedback and comments. On behalf of all the authors I enclose our replies to the various comments.

Best regards

Abdulhakim Ibrahim

Department of Vascular and Endovascular Surgery

University Hospital Münster

Albert-Schweitzer-Campus 1
48149 Münster

Germany

Tel:+49-251-8345788
E-Mail [email protected]

Reviewer 1

Comments and Suggestions for Authors

The authors of the manuscript aimed to assess the perioperative risk factors for SCI in patients who underwent aortic repair with FET technique 12 after acute aortic Stanford A dissection. They included 31 pts of which 4 suffered from SCI. They concluded that postop acute kidney injury, reduction of patent intercostal arteries after surgery and extension of false lumen thrombosis up to and beyond zone 5 may play a significant role in the development of clinically relevant SCI after FET.

Major remarks:

  1. I am pretty sure the study is heavily underpowered thus it’s almost impossible to find any relevant predictors of SCI in such small pts group. This is also the major limitation of the current study.

Thank you for your comment. The SCI is fortunately a rare complication of FET and FET has restricted specific indications, which limit the amount of cases our study analysis. We are aware that we have a limited number of patients in this study (like most single center studies using frozen elephant trunk for aortic dissection) and have stated this in our limitation.

To our knowledge, other studies, which analyse the specific risk factors for SCI after FET for acute aortic dissection, are small cohorts with limited size sampling, as example:

  • Kozlov BN, Panfilov DS, Ponomarenko IV, et al. The risk of spinal cord injury during the frozen elephant trunk procedure in acute aortic dissection. Interact Cardiovasc Thorac Surg. 2018;26(6):972-976. doi:10.1093/icvts/ivx432

or

  • Hohri Y, Yamasaki T, Matsuzaki Y, Hiramatsu T. Early and mid-term outcome of frozen elephant trunk using spinal cord protective perfusion strategy for acute type A aortic dissection. Gen Thorac Cardiovasc Surg. 2020;68(10):1119-1127. doi:10.1007/s11748-020-01328-z)

Kozlov et al. describe their findings in a group of 37 patients and the article of Hohri Y et al. published analyses 33 cases.

Our study analyses 31 patients, among which 4 cases of SCI occurred.

  1. In such low sample sized studies the authors should emphasize observational form of the study.

Thank you for your comment. We emphasized the observational form of our study in our text and in the article´s title

Change in text:

Title: Analysis of spinal ischemia after frozen elephant trunk for acute aortic dissection: an observational single center study

Abstract, background: Abstract: Background: This observational study aims to evaluate the perioperative risk factors for spinal cord ischemia (SCI) in patients who underwent aortic repair with frozen elephant trunk technique (FET) after acute aortic Stanford A dissection

  1. Moreover, the statistics reminds comparison of apples with plums and nothing more.
  2. Ask your statistician why UV regression and MV regression could not be performed.

Thank you for your important comments, which should definitely be integrated into this study. We took the known risk factors of SCI after endovascular therapy of the aorta using TEVAR (the risk factors for SCI after FET are not known) and compared preoperative, intraoperative and postoperative risk factors (which could cause tissue damage in the spinal cord, such as hypotension, hypoxemia, low Hb.....etc) in our groups and excluded these risk factors for our patient cohort. The statistics are burdened due to the number of patients and therefore we have dispensed with a UV and MV analysis.

Minor remarks:

  1. In the methods section please specify whether the study was prospective or retrospective

Thank you for your comment. We will specify it in the methods section.

Change in text (Methods): From May 2015 to April 2019, 31 Patients underwent aortic arch replacement with frozen elephant trunk (FET) technique using Thoraflex Hybrid prosthesis. Spinal ischemia was observed in 4 patients. Risk factors for SCI in this group of 31 patients were evaluated in a retrospective analysis.

  1. Did you obtain IERB approval?

Yes, we did. We specified it under:

Institutional Review Board Statement: The study was conducted in accordance with the Declarationof Helsinki, and approved by Medicine Faculty Münster (protocol number 2020-126-f-S).

  1. What was the duration of follow-up?

Postoperative spinal ischemia after FET is known as an early postoperative neurologic complication. Symptom onset after the first postoperative week is extremely rare. Therefore, active search of SCI symptoms was performed  during in-hospital care of all patients. This occurred until the seventh postoperative day; no patient showed  SCI-symptoms after the second postoperative day.

  1. The best feature of the current manuscript are the figures. Congratulations.

Thank you for your comment.

  1. English language requires minor editing.

Thank you for your comment. We revised and improved the text.

Reviewer 2 Report

The submission by Lomonaco Cuellar, et al provides an intersting analysis of the factors associated with spinal ischemia after surgery for aortic dissection using the frozen elephant trunk technique. The study relies on the experience with 4 patients with spinal ischemia among 31 operated on with this approach. The study is well written, easy to understand (with drawings detailibg the surgical procedure and its complications). The authors are easy to read, including for nonspecialized physicians. A few minor suggestions only :

Introduction : Please rephrase « Involving the 31 entire aorta, the residual false lumen of the downstream aorta plays an essential role in 32 determination of prognosis »

R).

Also, in the discussion section, a few words on spinal ischemia as reported by Hori Y, et al, J Thorac Cardiovasc Surg, 2020, 159 : 1119, and related editorial by Liacopoulos 2020 ; 159 : 1199

Reference section : add a recent reference on a modified approach (Berdajs, J Thorac Cardiovasc Surg 2022 ; 163 : 1754 and editorial comment by Hobbs

Author Response

Muenster, 31/10/2022

Editorial Office

MDPI: Diagnostics

Object: ID: diagnostics- 1987419

Analysis of spinal ischemia after frozen elephant trunk for acute aortic dissection: single center study

Dear editors and reviewers,

Thank you for your feedback and comments. On behalf of all the authors I enclose our replies to the various comments.

Best regards

Abdulhakim Ibrahim

Department of Vascular and Endovascular Surgery

University Hospital Münster

Albert-Schweitzer-Campus 1
48149 Münster

Germany

Tel:+49-251-8345788
E-Mail [email protected]

Reviewer 2

The submission by Lomonaco Cuellar, et al provides an intersting analysis of the factors associated with spinal ischemia after surgery for aortic dissection using the frozen elephant trunk technique. The study relies on the experience with 4 patients with spinal ischemia among 31 operated on with this approach. The study is well written, easy to understand (with drawings detailibg the surgical procedure and its complications). The authors are easy to read, including for nonspecialized physicians. A few minor suggestions only :

Introduction : Please rephrase « Involving the entire aorta, the residual false lumen of the downstream aorta plays an essential role in determination of prognosis »

Thank you for your comment. We rephrased the section.

Change in the text (introduction):

(Page 1 , line: 31 ) The residual false lumen may involve the entire downstream aorta and plays an essential role in determination of prognosis.

Also, in the discussion section, a few words on spinal ischemia as reported by Hori Y, et al, J Thorac Cardiovasc Surg, 2020, 159 : 1119, and related editorial by Liacopoulos 2020 ; 159 : 1199

Reference section : add a recent reference on a modified approach (Berdajs, J Thorac Cardiovasc Surg 2022 ; 163 : 1754 and editorial comment by Hobbs

Thank you for your comment. We improved the discussion section based on your suggestion and referenced the article in our paper.

Change in the text (discussion):

(Page 8 , lines 208-210):

Furthermore, FET for acute type A aortic dissection was associated with postoperative SCI rate of 4%. In their study of 31 patients, Hori et al. reported 4 case of postoperative SCI  (12.9%) [3].

(Page 8 , lines 211-218):

The deliberate use of FET has been discouraged by some surgeons because of the increas-ing evidence suggesting a higher SCI rate after FET than after conventional ascending aorta and hemiarch replacement, and even in comparison to the conventional elephant trunk technique. The multicenter ARCH registry found no statistical difference between the rates of SCI after FET and after conventional aortic repair. However, a retrospective ARCH registry analysis by Liakopoulos found several limitations in this registry analysis, proposing that more robust evidence is necessary to determine which subgroup of pa-tients would benefit most from the FET technique [13].

In our discussion section, Mr Hori´s findings were also already mentioned (Page 8 , lines 251-253):

Our study also suggests that postoperative acute kidney injury may present a risk factor for the postoperative development of SCI. Renal dysfunction has been previously reported as a risk factor for SCI after aortic repair [3, 18].

Change in the text (discussion):

(Page 8 , lines 196-205):

A potential shortcoming of FET is the necessity of a complex arch replacement; a hemi-arch replacement might be a more adequate approach in selected cases [12]. 

An alternative hybrid method for treating acute Type A Dissection is the modified frozen elephant trunk (mFET). This procedure leaves the native aortic arch is left in situ with the remaining dissection; the ascending aorta is replaced in open surgery and the descending aorta is repaired with an endovascular stentgraft. In a propensity-based anal-ysis, Berdajs et al. compared mFET to isolated ascending aorta and hemiarch replacement. Their findings regarding midterm-outcome favored mFET [14]. However, this study did not compare outcomes of mFET and FET [12].   

Round 2

Reviewer 1 Report

Thank you. I have no further comments.